# Characterization of Chitin Synthase A cDNA from *Diaphorina citri* (Hemiptera: Liviidae) and Its Response to Diflubenzuron

**DOI:** 10.3390/insects13080728

**Published:** 2022-08-15

**Authors:** Cong Zhang, Wenfeng Hu, Zhuo Yu, Xian Liu, Jing Wang, Tianrong Xin, Zhiwen Zou, Bin Xia

**Affiliations:** School of Life Sciences, Nanchang University, Nanchang 330031, China

**Keywords:** *Diaphorina citri*, chitin, diflubenzuron, transcription level, RNA interference

## Abstract

**Simple Summary:**

The transcriptional level of chitin synthase A (*DcCHSA*) was investigated in *Diaphorina citri* Kuwayama at various developmental stages and tissues in this work. The survival and molting of *D. citri* nymphs were negatively impacted by high concentrations of DFB, which could also cause *DcCHSA* to express itself significantly. We also looked into how CHS knockout affected nymph development and eclosion. The findings revealed that *DcCHSA* was critical for the eclosion and molting of *D. citri* nymphs. This study provides a solid theoretical foundation for *D. citri* management in China.

**Abstract:**

*Diaphorina citri* Kuwayama is the vector of HLB and one of the most common pests in citrus orchards in southern China. One of the most significant genes in *D. citri*’s growth and development is the chitin synthase gene. In this study, the CHS gene (*DcCHSA*) of *D. citri* was cloned and analyzed by bioinformatics. According to RT-qPCR findings, *DcCHSA* was expressed at many growth processes of *D. citri*, with the greatest influence in the fifth-instar nymph. The molting failure rate and mortality of *D. citri* rose as DFB concentration increased in this research, as did the expression level of *DcCHSA*. Feeding on *DcCHSA* caused a large drop in target gene expression, affected nymph molting, caused failure or even death in freshly eclosion adults, increased mortality, and reduced the molting success rate over time. These findings showed that *DcCHSA* was involved in nymph to adult development and may aid in the identification of molecular targets for *D. citri* regulation. It provided new ideas for further control of *D. citri*.

## 1. Introduction

In China, citrus fruits are mainly *Citrus reticulata Blanco*, which is the most widely planted citrus fruit in Asia [1]. However, citrus crops are particularly vulnerable to insects, which can cause many diseases [2,3]. Huanglongbing (HLB), sometimes known as “citrus cancer,” is one of the most dangerous illnesses. *Diaphorina citri* Kuwayama (Hemiptera: Liviidae) is the vector of HLB and one of the most common pests in citrus orchards in southern China. By ingesting the juice from Citrus phloem, they disseminate pathogens (*Candidatus* Liberibacter asiaticus (CLas)) [4,5]. *D. citri* are currently found in 10 Chinese provinces, including Jiangxi, Guangdong, Guangxi, Yunnan, Sichuan, and Hainan [6]. HLB has spread more quickly and with a wider range of damage in recent years as a result of the population of *D. citri* considerably increasing and its geographic distribution moving further north each year due to rising winter temperatures [7]. Wang et al. combined the ecological niche modeling software MaxEnt with ArcGIS to predict the potential geographic distribution of *D. citri* in China. The results displayed that *D. citri* had the possibility of further spreading in China in the future [8].

During growth and development, insects, such as *D. citri*, must molt on a regular basis and replace their stiff exoskeleton [9]. The cuticle of insects’ exoskeleton is vital to their survival because it can provide strength, defend against enemies, and prevent water loss [10]. Most of the cuticles of insects are made of chitin [11]. Insect tissues and organs such as the peritrophic membrane, trachea, and epidermis all contain chitin [12]. Chitin synthase (CHS) and chitin-degrading enzymes are required for chitin synthesis [13]. *CHSA* and *CHSB* are the two types of chitin synthase found in most insects (also called *CHS1* and *CHS2*) [14]. Class A enzymes produce chitin in the epidermis, while class B enzymes are responsible for the synthesis of chitin in the intestinal epithelium [15]. Currently, only *CHSA* has been discovered in Hemiptera insects (*Nilaparvata lugens* and *Aphis glycines*) [11,16]. However, whether two chitin synthase genes exist in *D. citri* is one of the goals of this study. As with other Hemiptera insects, *D. citri* has only one CHS gene [17].

Chitin has not been found in vertebrates, but only in arthropods, fungi, and nematodes, so it can be used as a target for insecticides [18]. The first commercial insecticide that works by preventing the formation of chitin in insects was called diflubenzuron (DFB) [19]. It is an insecticide made of benzoyl urea (BFU) that interacts with CHS to prevent the production of chitin, preventing insect molting and causing other physiological changes [20,21]. DFB has been widely used in the control of molting and metamorphosis of various pests [22]. The effect of DFB on the growth and development of citrus psyllids has been reported, e.g., according to Tiwar et al. [23], DFB can successfully prevent the emergence of adults. DFB has not, however, been employed to manage *D. citri* in China. Giving a potential theoretical foundation for DFB application in China is one of the goals of this work.

Both functional genomic research on insects and the management of pest insects can benefit from the use of RNA interference [24,25]. Insecticidal gene silencing by RNA interference (RNAi) is one of the biotechnological advancements expected to revolutionize pest control [24]. The RNA interference pathway begins when Dicer cuts dsRNA into siRNAs and then targets the homologous mRNAs for destruction [26]. RNAi techniques have identified several potential control target genes in *D. citri*, such as carboxylesterase [27], arginine kinase [28], and muscle protein-20 [29]. However, some potential target genes show varying degrees of sensitivity to RNAi [30]. The findings revealed that genes with greater levels of expression were more susceptible to RNAi [30]. Therefore, we need to examine how much CHS is expressed at different phases of *D. citri* growth to determine the optimal timing of RNAi. Galdeano et al. [31] studied RNAi technology with the CHS gene of *D. citri* as the target. Lu et al. [32] knocked-down the CHS gene of *D. citri* by RNAi technology and only found abnormal wing development in adults. It is not clear what specific effects the reduction of CHS gene transcription will have on the growth and development of nymphs. It is necessary to design more effective dsRNA sequences from the complete sequence of the CHS gene to study the important role of CHS in the growth and development of *D. citri*.

In this study, we identified a full-length cDNA encoding *CHSA* and its complete open reading frame (ORF). Based on the amino acid sequence alignment of several insect chitin synthases, a phylogenetic tree was generated. Quantitative real-time PCR (qPCR) was used to explore *DcCHSA* expression levels at various life stages. The mortality of fifth-instar nymphs significantly increased after being exposed to DFB. We also discovered that dsRNA silencing resulted in a significant reduction in *DcCHSA* relative expression and nymph molting rate, an increase in mortality, and the failure of adult wing expansion. By studying *DcCHSA*, we hope to provide a new basis for the field control of *D. citri* in China.

## 2. Materials and Methods

### 2.1. Insect

Insects were collected from *Murraya exotica* in Quanzhou, Fujian Province, China in 2018. *D. citri* were reared in mesh cages (60 cm × 60 cm × 90 cm) on *Murraya exotica* with 27 ± 1 °C, 70 ± 5% relative humidity, and 14:10 h light:dark photoperiods.

### 2.2. Leaf-Dip Bioassay

DFB was a prepared insecticide used in bioassays (Aladdin^®^, Aladdin Industrial Corporation, Ontario, CA, USA). We employed a modified leaf-dip bioassay that was based on a prior methodology [33]. As a stock solution, 150 mg of diflubenzuron was weighed, 150 mL of acetone was added, and 1 mg/mL of DFB solution was made. Then, five concentrations (5, 50, 100 mg/L, 200 mg/L, and 500 mg/L) were made with 0.01% surfactant Triton X-100 (Sangon Biotechnology, Shanghai, China). The control was treated with 0.01% surfactant Triton X-100. Isolated fresh leaves (*M. exotica*) were immersed in DFB solution for 5 min and air-dried for 1 h before 30 5th-instar nymphs were transferred to each leaf. The test insects were maintained at 27 °C with a 14:10 (L:D) photoperiod in a climate-controlled environment. The parameters including the regression equation, median lethal concentration, and correlation coefficient were calculated by SPSS 24.0. According to the virulence equation, three concentrations (5, 100, and 500 mg/L) were selected to evaluate the effect of DFB on the 5th-instar nymphs. At 48 h after the DFB exposure, the number of deaths and molts were counted, and the mortality and abortion molting rates were calculated. The mortality statistics also included the individuals who died due to abortive molting. Dead and surviving individuals were also included in the abortive molting rates. Three replicates were used for each experiment.

### 2.3. Total RNA Extraction and cDNA Synthesis

Extraction of total RNA from *D. citri* was conducted with a kit (Eastep^®^ Super Kit, Shanghai Promega Biological Products Co., Ltd., Shanghai, China) to create the *DcCHSA* cDNA template. A spectrophotometer (NanoDrop2000; Thermo Fisher Scientific, Waltham, MA, USA) was used to measure the concentration of RNA, and the quality of the RNA was determined using 1% agarose gel electrophoresis. The PrimeScriptTMII 1st Strand cDNA Amplification Kit (TaKaRa Bio, Dalian, China) was used to make the first-strand cDNA, and the RACE-PCRs were conducted with a SMARTer™ RACE cDNA Amplification Kit (TaKaRa Bio, Dalian, China). The generated cDNA was maintained at −20 °C after reverse transcription for future use.

### 2.4. Cloning and Sequencing of DcCHSA

We obtained some CHS gene sequences from the transcriptome data of *D. citri*, which were highly similar to the CHS Gene previously studied (GenBank: XP_017303059). Degenerate primers and specific primers were designed using Primers 5.0. These primers were used to amplify unidentified sequences and validate obtained sequences (Table 1). Rapid amplification of cDNA ends (RACE) PCR was performed using the SMARTer^®^ RACE 5′/3′ Kit (Takara Biomedical Technology (Beijing) Co., Ltd., Beijing, China ) by the standard protocol. The cDNA end clones for the *DcCHSA* genes and the 25 µL PCR reaction system included: 12.5 µL of 2 × Tag PCR Master Mix (TaKaRa Bio, Kusatsu, Japan), 9.5 µL of sterile H2O, 1.0 µL each of forward and reverse primers (10 µmol·L^−1^), and 2 µL of the sample cDNA template (100 ng). The PCR results were detected by electrophoresis (1% agarose gel) and the correct fragment was recovered from the agarose gel using a gel extraction kit (SanPrep Column DNA Gel Extraction Kit, Sangon Biotechnology, Shanghai, China). The recovered products were cloned into the 007VS vector (Tsingke Biotechnology Co., Ltd., Beijing, China) and then transformed into competent Escherichia coli cells (Tsingke Biotechnology Co., Ltd., Beijing, China). The monoclonal colonies were selected for PCR verification, and the positive clones were sequenced (Sangon Biotechnology, Shanghai, China).

### 2.5. Real-Time Quantitative PCR (RT-qPCR)

To determine the best time of RNAi, *D. citri* at various times (1st–5th-instar nymphs and adults) were collected for RT-qPCR detection. In addition, to obtain tissue specificity, the heads, wings, integuments, mycetomes, guts (containing foregut, midgut, and hindgut), female ovaries, and male testis of adults were dissected. Under a Leica S9i (Leica Microsystems GmbH, Wetzlar, Germany) stereomicroscope, the pests were dissected in phosphate-buffered saline (PBS) (Sangon Biotech Shanghai Co., Ltd., Shanghai, China). The isolated tissues were put into Trizol (Invitrogen, New York, NY, USA) and stored at −80 °C until RNA was extracted. According to Trizol’s instructions, total extract RNA was from tissues and insects of different ages, and the synthesis method of cDNA was as described above. For nymphs during various stages of growth, each sample was made up of 20–30 pests and was repeated three times. For different tissues, each sample contained 50–60 pests and was repeated 3 times.

RT-PCR primers for *DcCHSA* are listed in Table 1. The RT-qPCR was run using a StepOnePlusTM Real-Time PCR system (Thermo Fisher Scientific, Singapore). β-Actin (GenBank: DQ675553) and α-tubulin (GenBank: DQ675550) were used as internal reference genes. The reaction system was: 10 µL of TB green Mix, 7.8 µL of ddH2O, 1.0 µL of cDNA, 0.4 µL of Rox dye, and 0.4 µL of each primer. The reaction procedure was: 95 °C for 3 min and 40 cycles at 95 °C for 10 s and 60 °C for 20 s. Three replicates were performed for each reaction. The 2^−ΔΔCt^ was used to determine the relative expression levels of *DcCHSA* in each sample [34].

### 2.6. Preparation of dsRNA and Feeding

To investigate the biological roles of *DcCHSA* in *D. citri*, RNAi was used. The most distinctive *DcCHSA* nucleotide regions were chosen for particular dsRNA production. According to the manufacturer’s instructions, *DcCHSA* was engineered to produce dsRNA using the T7 High Yield Transcription Kit (Vazyme Biotech Co., Ltd., Nanjing, China). Table 1 lists the primers used to manufacture dsRNA. ds*GFP* was used in the control group. The size of the dsRNA products was confirmed by electrophoresis on a 1% agarose gel and the final concentration of dsRNA was 300 ng/μL. A total of 50 newly emerged 5th-instar nymphs were used in dsRNA treatment. A 20% (*w*:*v*) sucrose artificial diet was combined with ds*DcCHSA* at a final concentration of 500 ng/L. Nymphs were placed on stretchable parafilm with artificial feed (200 μL) between the two membranes. All live insects were collected after 48 h to extract total RNA and synthesize cDNA. RT-qPCR was used to assess the effect of ds*DcCHSA* on gene expression. Each experiment included three biological replicates.

### 2.7. Bioinformatic and Phylogenetic Analyses and Statistical Analysis

The sequence similarities were analyzed using blast programs in the NCBI databases (available online: http://www.ncbi.nlm.nih.gov/, accessed on 21 August 2021) and open reading frames were predicted at ORF finder (available online: http://www.ncbi.nlm.nih.gov/gorf/gorf.html, accessed on 14 October 2021). The molecular weight (MW) and isoelectric point (pI) of the deduced protein sequences were obtained using the ExPASy portal (available online: http://web.expasy.org/compute_pi/, accessed on 14 October 2021). The N-glycosylation sites were predicted by the NetNGyc 1.0 Server (http://www.cbs.dtu.dk/services/NetNGyc/), and the transmembrane helices were analyzed using TMHMM V.2.0 (http://www.cbs.dtu.dk/services/TMHMM-2.0/, accessed on 14 October 2021). The NetPhos 3.1 Server (http://www.cbs.dtu.dk/services/NetPhos/, accessed on 14 October 2021) predicted the phosphorylation site of the amino acid sequence and predicted the spatial structure. MEGA 6.0 and the neighbor-joining approach were used to create the phylogenetic tree, with bootstrap values determined on 1000 iterations [35]. Table 2 lists the GenBank IDs of the *CHS* utilized to create the tree, which came from 21 insects.

### 2.8. Statistical Analysis

The data were summarized using the mean ± SE for all datasets. A one-way analysis of variance (ANOVA) was performed using SPSS 26.0. The Student–Newman–Keuls (S–N–K) test was used to evaluate mean differences for multiple comparisons. All experiments were performed with three biological replicates.

## 3. Results

### 3.1. Cloning and Sequence Analysis of DcCHSA

We identified a full-length 4677 bp long cDNA encoding for a CHS gene (*DcCHSA*) (Figure 1). The full-length confirmation experiments showed that the complete cDNA sequence of *DcCHSA* contained an ORF of 4437 bp that encoded a 1478-amino-acid residue with a predicted molecular mass of 169.74 kDa and an isoelectric point of 6.35. This result is different from previous studies [32]. The possible reason is that the previous studies did not obtain a complete gene sequence, or the previous sequences have sequence deletion and duplication regions [17].

Figure 1 shows the nucleotide and projected amino acid sequences of the *DcCHSA*. Scanning of the deduced amino acid sequence of *DcCHSA* at TMHMM Server V. 2.0 (predicted 15 hydrophilic, membrane-spanning helices. In addition, the *DcCHSA* protein was predicted to contain 11 N-glycosylation sites. The putative catalytic domain of *DcCHSA* was 339 amino-acid-residues long and contained two signature motifs, i.e., EDR and QRRRW. The *DcCHSA* gene had a total of 20 amino acids, of which Leu was most common (159), and Trp was least common (28). The NetPhos 3.1 Server was used to predict the phosphorylation site of this protein, which had 234 phosphorylation sites.

### 3.2. Phylogenetic Analysis of DcCHSA

Utilizing MEGA (version 6.0), a phylogenetic tree was created using the entire amino acid sequence of this study and those of other insects (Figure 2). CHS proteins were divided into two branches, A and B. *DcCHSA* in this study belonged to *CHSA*. Interestingly, *D. citri* and the Aphididae family insects gathered into one branch, and their bootstrap value was high, reaching 99. In sum, phylogenetic analysis showed that *DcCHSA* belonged to orthologs of several hemipteran species including *Aphis citricidus* and *Aphis glycines*.

### 3.3. Gene Expression Profiles of DcCHSA

*DcCHSA* transcription values were observed using RT-qPCR at different growth stages (Figure 3A). The transcriptional levels of *DcCHSA* showed differences from the egg to the adult. The expression of *DcCHSA* in a fifth-instar nymph was highest, 82.3 times that of the egg, 14.7 times that of the first-instar nymph, 18.8 times that of a second-instar nymph, 5.9 times that of the third-instar nymph, 3.6 times that of the fourth-instar nymph, and 1.8 times that of the adult. The adult stage had the second greatest level of expressiveness. The *DcCHSA* was almost not expressed in the egg stage, but slightly decreased in the second-instar nymph stage, significantly increased in the fourth-instar nymph to the fifth-instar nymph, and significantly decreased in the emergence to the adult. The findings of first- to third-instar nymphs conflict with earlier research (Figure 3A). Examining the transcription level of *DcCHSA* in the nymph and adult stages, Lu et al. [32] found that the expression of *DcCHSA* is highest in the first- and second-instar nymphs and lowest in the third-instar nymphs. Tissue-specific results showed that the transcriptional level of *DcCHSA* was highest in the head and integument (Figure 3B). The expression in the gut, on the other hand, was much lower than in the head and integument. *DcCHSA* expression was much higher in the ovaries of female adults than in the testis of male adults, second only to the head and integument.

### 3.4. Effect of DFB on D. citri Survival and DcCHSA Expression Level

The results of the virulence test showed that the median lethal concentration for *D. citri* fifth-instar nymphs exposed to DFB was 140.9 mg/L (Table 3). With the increase in DFB concentration, the toxic effect of DFB on *D. citri* became more obvious. When treated with 500 mg/L of DFB, the cumulative mortality of the treatment (83%, including individuals with abortive molting) after 48 h was higher than those of the control and other concentrations (5 mg/L, 8%; 100 mg/L, 46%) (Figure 4A). Among 17% of the surviving individuals, 60% were those who failed to spread their wings. Similarly, at 48 h, the rate of abortive molting in *D. citri* treated with 500 mg/L of DFB (31%, including dead and living individuals) was greater than that of the control (1%), that treated with 5 mg/L of DFB (3%), and that treated with 100 mg/L of DFB (14%) (Figure 4B). In addition, as the concentration of DFB increased, the proportion of insects that could not completely molt and failed to spread wings increased. Even at 500 mg/L of DFB, blisters appeared on the left front side of the nymphs’ thoraxes (Figure 5B). All of these individuals with blisters could not emerge and died. The relative expression of *DcCHSA* increased significantly from 48 h after being induced by high DFB concentrations (Figure 4C).

### 3.5. DcCHSA RNAi Analysis

To explore the *DcCHSA* effects on the nymph to the adult transition of *D. citri*, insects were collected for feeding with ds*DcCHSA*. The fifth-instar nymphs with the greatest gene expression were chosen for RNAi based on the gene expression profile. After 48 h, the cumulative mortality in the ds*DcCHSA* group was 24%, 4% in the ds*GFP* group, 2% in the control, and significantly higher in the ds*DcCHSA* group (Figure 6A). After 48 h of feeding, the expression of the ds*DcCHSA* decreased by 38.26% compared to the ds*GFP* and 44.91% compared to the control (Figure 6B). Compared with the control, nymphs had partial molting and even molting failure until death (Figure 7B). The freshly emerged adults failed to spread their wings, and the abdomen was unable to shed the old epidermis (Figure 7C). These findings revealed that inhibiting *CHSA* expression has a major impact on the nymph-to-adult metamorphosis.

## 4. Discussion

In this study, the CHS cDNA (*DcCHSA*) sequence of *D. citri* was successfully cloned. Sequence alignment analysis revealed that *DcCHSA* contains two chitin synthase tag sequences, EDR and QRRRW. This indicates that *DcCHSA* has the structural characteristics of chitin synthase and may have similar physiological functions as *CHSA*. The CHS protein of insects usually contains 15 transmembrane helices [10]. The CHS protein obtained in this study also has 15 transmembrane helices. This was the same as the annotation results of Miller et al. [17]. Our findings differ from those of Lu et al. [32], in part, because of missing sequences and repetitive areas in the previously predicted *D. citri* CHS (xp017303059). The phylogenetic analysis also showed that *DcCHSA* had a higher genetic relationship with the *CHS1* gene of brown citrus aphid [36]. These results indicate that the *DcCHSA* obtained in this study belongs to *CHS1*. Generally, insects have two CHS genes [13,37]. Interestingly, there was just one CHS gene, according to several studies [11,36], *CHSA*, in Hemiptera. This study also supports the results that Hemiptera only contains *CHSA*. In many insects, the midgut epithelium is lined with an acellular, semipermeable structure referred to as the peritrophic matrix (PM) [13,38]. However, the research shows that there is a lack of PM in Hemiptera [11]. This may be the reason that there is only the *DcCHSA* gene in *D. citri*. Additionally, this study did not suggest alternative splicing variations in *DcCHSA*, which is consistent with aphid investigations [11,36]. In Hemiptera insects such as *Nilaparvata lugens*, *Laodelphax striatellus*, and *Sogatella furcifera*, alternate splicing variants do exist [16,39]. In addition, mutations in some loci of *CHS* genes can enhance insect insecticide resistance [40]. Therefore, it is necessary to further study *DcCHSA*.

Developmental expression of *DcCHSA* from eggs to adults was determined by RT-qPCR. Our findings demonstrated that *DcCHSA* expression was identified in all physiological stages, which is consistent with the findings of *Acyrthosiphon pisum* [41]. *DcCHSA* expression, on the other hand, was lowest in eggs and greatest in fifth-instar nymph. The lowest expression level in eggs was opposite to those of the *Aphis glycines Ay CHS* [11] and *PcCHS1* gene of *Panonychus citri* [33], but it was consistent with that of the *Locusta migratoria manilensis LmCHS1* gene [42]. The development and growth of insects are affected by periodic molting, during which the insects expand their exoskeleton to adapt to internal growth. During this process, insects need to significantly increase exoskeletal chitin to form new cuticles [42]. When the *D. citri* develops to the fifth-instar nymph, its body size increases sharply, and a large amount of chitin needs to be synthesized to meet its own needs. The fifth-instar nymph is also considered the critical period for the *D. citri* to move from the nymph to the adult [43]. This may be the reason for the highest expression of *DcCHSA* in the fifth-instar nymph stage. The results of tissue expression were similar to those of other insects. *CHSA* is exclusively found in epidermal cells originating from ectoderm, such as the epidermis, trachea, and salivary gland, and is responsible for the formation of chitin in these tissues, according to prior research [14,44]. *CHSA* also exhibits a high level of expression in the ovaries, indicating that it is involved in ovarian development and insect reproduction [45,46]. As a consequence of the findings, thus far, additional research is required to fully comprehend the role of *DcCHSA* in *D. citri*.

The incidence of molting failure and mortality increased as the concentration of DFB increased, as did the expression level of *DcCHSA*, which is consistent with earlier findings [11,36]. The molting failure rate was greatest when the concentration of DFB was 500 mg/L, and a distinct phenotype formed. DFB was the first commercial insecticide that acts by inhibiting chitin synthesis in insects [19]. Insects exposed to this chitin synthesis inhibitor may acquire aberrant cuticle forms, abnormal procuticle depositions, and abortive molting [47]. Blisters developed on the thorax and abdomen of nymphs treated with a high concentration of DFB, which may be because the stratum corneum cannot tolerate the increased turgor pressure, leading to inadequate muscular support during molting [48]. However, DFB did not inhibit the expression of *DcCHSA*, which may not be the target of DFB. DFB was formerly thought to lower chitin content by decreasing chitin synthase activity [19]. The increased level of CHS expression can also indicate the existence of a feedback regulation mechanism to compensate for low enzyme content [33]. However, upregulation or mutation of CHS expression can improve insect resistance [40,49]. As a result, we should pay more attention to the *DcCHSA* molecular mechanisms in drug resistance in the future.

RNAi has been widely used in studies to understand how genes operate, and it has shown great promise for the creation of fresh pest control strategies [50]. We performed RNAi tests on the fifth-instar nymphs by feeding them ds*DcCHSA*. The dsRNA-mediated *DcCHSA* silencing produced a considerable drop in *DcCHSA* expression during the nymph to adult metamorphosis, affecting nymph molting and causing death. *Toxoptera citricida* and *Locusta migratoria manilensis* had the same outcomes when CHS was silenced [36,42]. Similarly, freshly emerging adults following RNAi were unable to stretch their wings successfully, which was comparable to a study of *Glyphodes pyloalis* [51]. This suggests that *DcCHSA* was involved in the synthesis of chitin in the exoskeleton of *D. citri*, and that it may be a crucial gene for the nymph to adult development, as well as for wing expansion. However, more research is needed to demonstrate how *DcCHSA* knockdown impacts other stages of *D. citri* development, including the transition from second- to third-instar nymphs.

## 5. Conclusions

In summary, we identified a complete sequence of the *CHS* gene in *D. citr*. RT-qPCR showed that the relative expression of *DcCHSA* was highest in the fifth-instar nymph. When exposed to DFB, the molting failure rate and mortality rate of *D. citri* increased with the increase in DFB concentration, and the expression of *DcCHSA* also increased. Exposure to ds*DcCHSA* by ingestion resulted in a significant decline in target gene expression and also affected nymph molting, leading to increased mortality and reducing molting success over time. These results further deepened our understanding of the biological functions of the *DcCHSA* gene in *D. citri* and provided new ideas for control.

## Figures and Tables

**Figure 1 insects-13-00728-f001:**
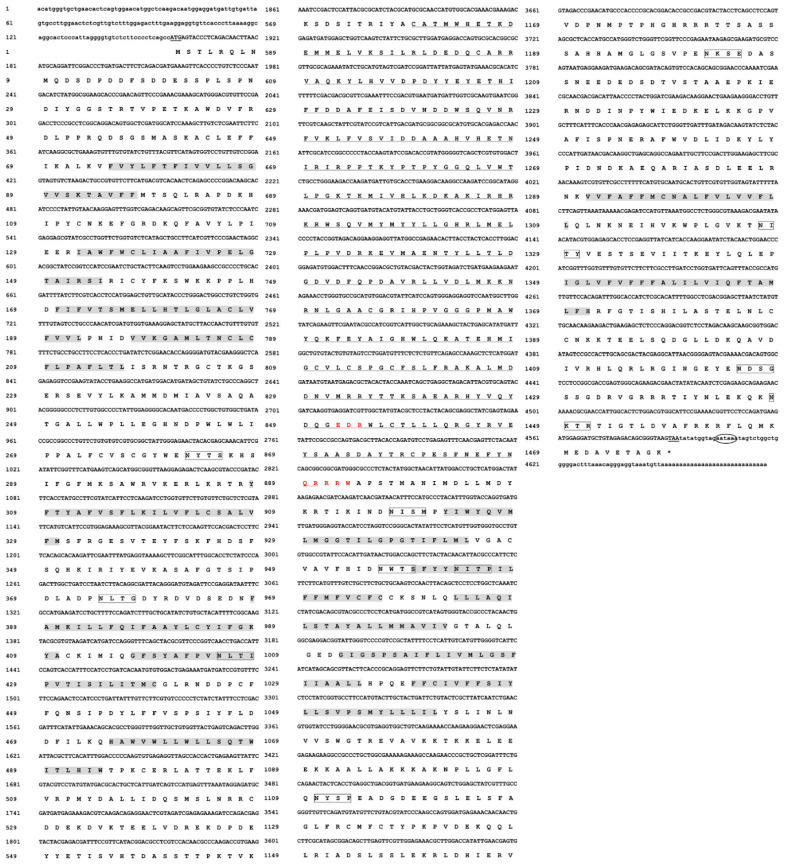
Nucleotide and deduced amino acid sequences of *DcCHSA* cDNA from *D. citri*. The start codon (ATG) and the stop codon (TAA) are indicated with black underlines, and the putative polyadenylation signal (AATAAA) in bold with a black ellipse. The 15 hydrophilic, membrane-spanning helices are indicated in gray font, the amino acid sequence of the putative catalytic domains is wavy-lined, and the chitin synthase signature motifs EDR (852–854 bp) and QRRRW (889–894 bp) are indicated in red font. Black boxes are used to indicate the 11 predicted N-glycosylation sites.

**Figure 2 insects-13-00728-f002:**
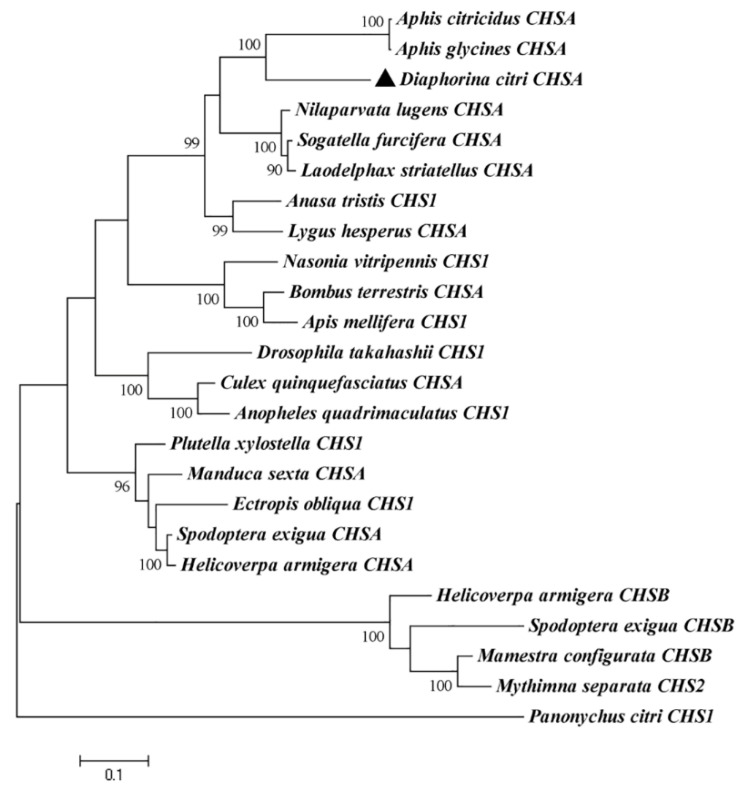
Phylogeny of insect chitin synthases. Phylogenetic relationships of *DcCHSA* in different insect species using the neighbor-joining method with a bootstrap value of 1000. Chitin synthases were from 21 insects.

**Figure 3 insects-13-00728-f003:**
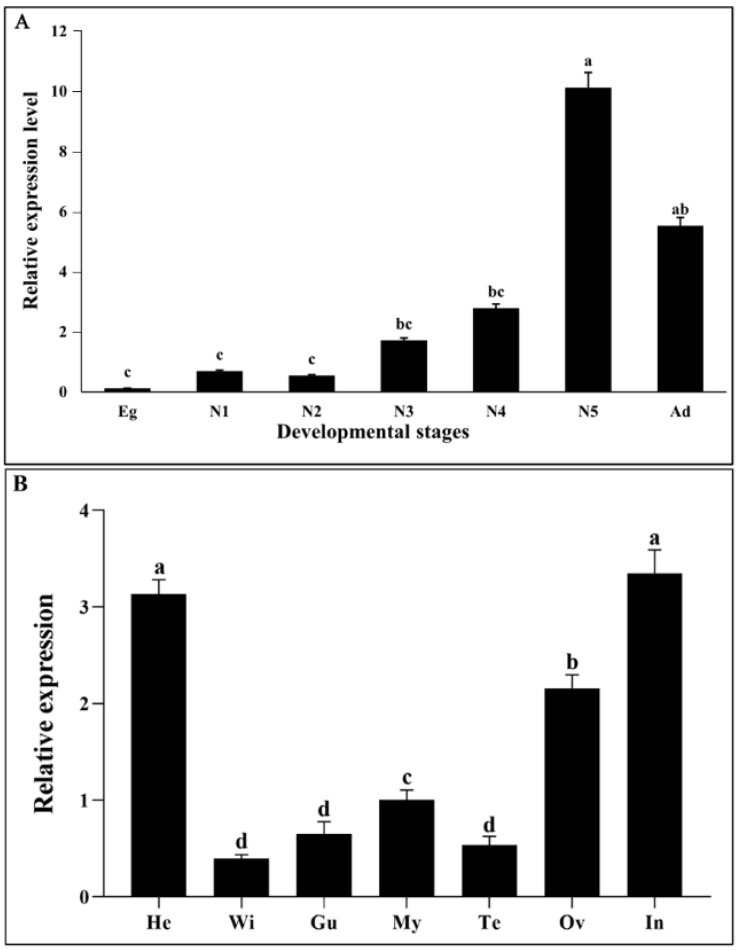
Relative expression levels of *DcCHSA* in different developmental stages and tissues of *D. citri*. (**A**) The expression of *DcCHSA* was highest in fifth-instar nymphs, followed by the adult stage. (**B**) Tissue-specific analysis showed that the transcription level of *DcCHSA* was highest in the head and integument, and the second highest was in the ovary. Different letters on the bars of the histogram indicate significant difference in gene expression compared to the treatment with lowest expression at *p* value < 0.05 (ANOVA). Eg, egg; N1, first-instar nymph; N2, second-instar nymph; N3, third-instar nymph; N4, fourth-instar nymph; N5, fifth-instar nymph; Ad, adult; He, Head; Wi, Wing; Gu, guts; My, Mycetome; Te, Testis; Ov, Ovary; In, Integument.

**Figure 4 insects-13-00728-f004:**
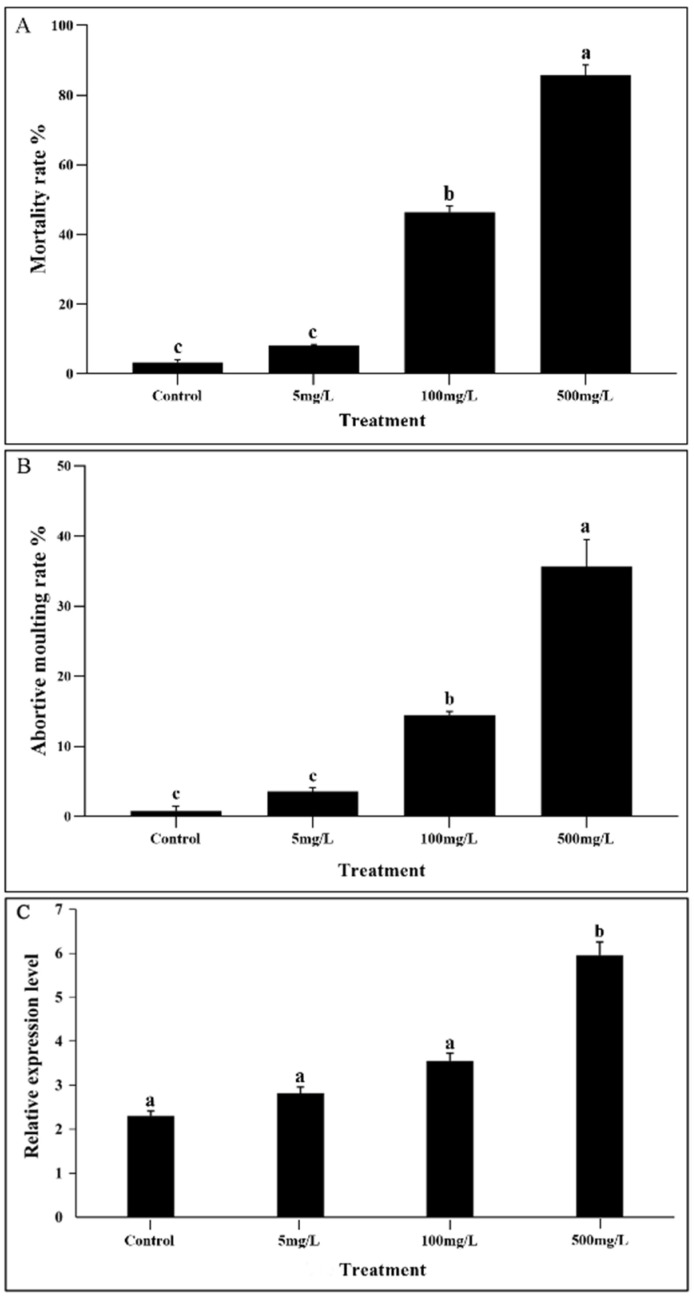
Effect of DFB on *D. citri* mortality, molting rate, and *DcCHSA* expression levels. (**A**) At 48 h, the cumulative mortality of *D. citri* in 500 mg/L DFB treatment was significantly higher than those in the control, 5 mg/L DFB, and 100 mg/L DFB treatments. (**B**) The abortive molting rate of *D. citri* treated with 500 mg/L of DFB at 48 h was significantly higher than that of the control group, that treated with 5 mg/L of DFB, and that treated with 100 mg/L of DFB. (**C**) The relative expression of *DcCHSA* in *D. citri* increased significantly at 48 h after being induced by low and high DFB concentrations. Different letters on the bars of the histogram indicate significant difference at *p* value < 0.05 (ANOVA).

**Figure 5 insects-13-00728-f005:**
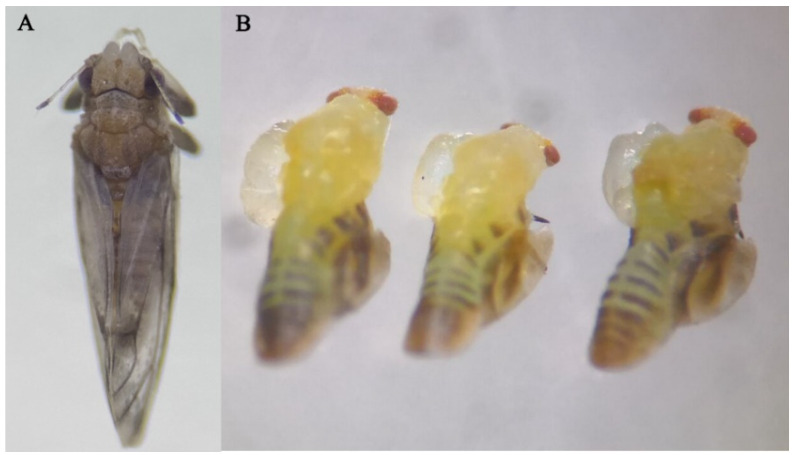
Exposed to 500 mg/L of DFB, *D. citri* nymphs showed a special phenotype. (**A**) The nymphs in the control (0.01% Triton X-100) can normally develop into adults. (**B**) Blisters appeared on the left front side of the nymph thorax in 500 mg/L of DFB.

**Figure 6 insects-13-00728-f006:**
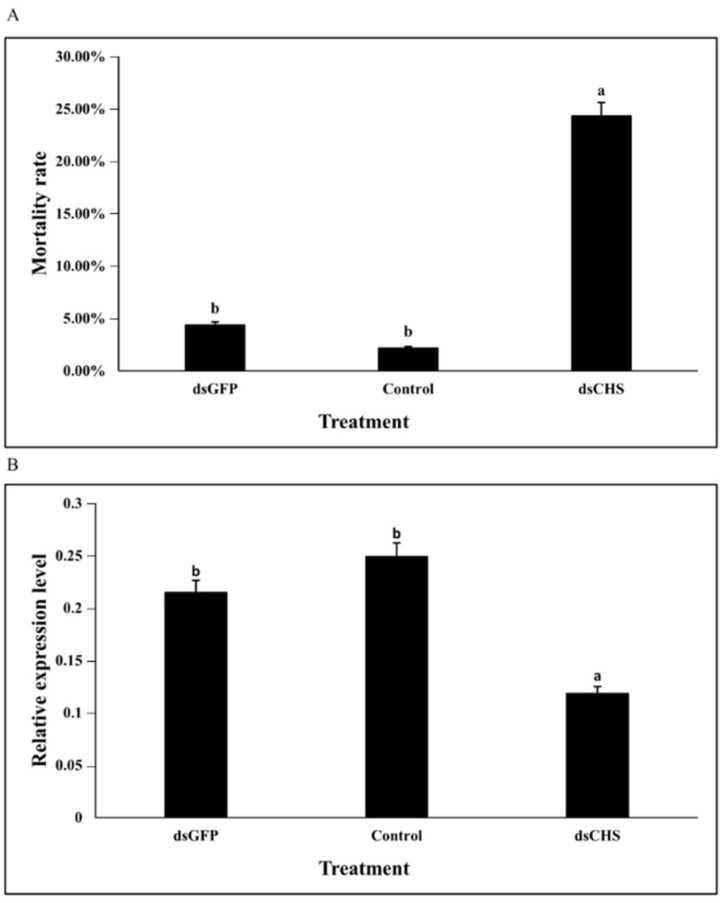
Effects on *D. citri* after RNAi of *DcCHSA.* (**A**) Cumulative mortality of *D. citri* in the treatment and control after RNAi of 48 h. After 48 h, the cumulative mortality of ds*DcCHSA* group was significantly higher than that of control group. (**B**) Relative expression levels of *DcCHSA* when *D. citri* was treated with ds*DcCHSA* and ds*GFP*. After feeding ds*DcCHSA* for 48 h, the transcript level of *DcCHSA* was significantly lower than that of the control group. Different letters on the bars of the histogram indicate significant difference at *p* value < 0.05 (ANOVA).

**Figure 7 insects-13-00728-f007:**
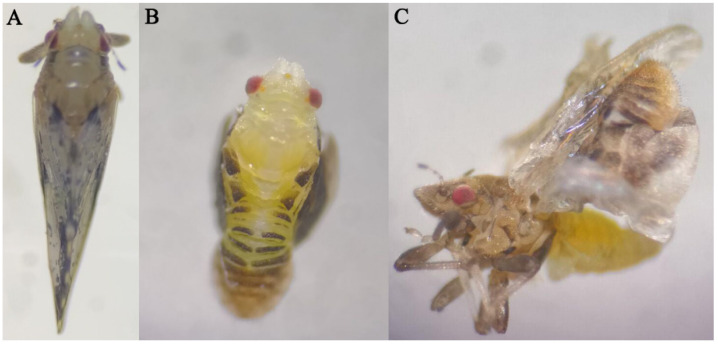
Fifth-instar nymphs treated with dsRNA. (**A**) Fifth-instar nymphs fed ds*GFP* can emerge normally. (**B**) After feeding dsRNA for 48 h, fifth-instar nymphs could not molt. (**C**) The freshly emerged adults failed to spread their wings, and the abdomen was unable to shed the old epidermis. A total of 30 fifth-instar nymphs were used for phenotypic observation after RNAi.

**Table 1 insects-13-00728-t001:** The *DcCHSA* expression analysis and cloning primers.

Experiments	Name	Sequences
PCR	*DcCHSA*5F	TWCTGCAATAAAGAYYTA
	*DcCHSA*5R	AKATAWRTGAARTAWCGWGT
	*DcCHSA*3F	ATGATGGAVATGWTNARSTSKAT
	*DcCHSA*3R	CATAAGRAATATHGTKCCHGGDC
	*DcCHSA*ZF	ACTACAACGAGCAAACATTCA
	*DcCHSA*ZR	GAGAATGGGCGTAATGTTGT
3′-RACE	*DcCHSA*F1	GACAAGCAAGCGGTGGACATA
	*DcCHSA*F2	AGGTCCGCGTCAACTCTTCTTCCATC
5′-RACE	*DcCHSA*R1	AGGCGGGAGGTCTCGGAACA
	*DcCHSA*R2	ATGGAGGTGACGAAGATAAA
3′/5′-RACE	NUP	AAGCAGTGGTATCAACGCAGAGT
	UPM (mix)	TAATACGACTCACTATAGGGCAAGCAGTGGTATCAACGCAGAGT
		CTAATACGACTCACTATAGGGC
RT-qPCR	*Dc*ActinF	CCATCTTGGCTTCTCTGTCTAC
	*Dc*ActinR	CATTTGCGGTGAACGATTCC
	*Dc*TubulinF	TTTCCAACACCACCGCTAT
	*Dc*TubulinR	AGGTCTTCCCTCGCCTCTG
	*DcCHSA*qF	CCACGACTCCTTCTCACAG
	*DcCHSA*qR	TCATGGCGAAATTATCCTC
RNAi	GFPiF	GCCAACACTTGTCACTACTT
	GFPiR	GGAGTATTTTGTTGATAATGGTCTG
	GFPdsF	taatacgactcactatagggGCCAACACTTGTCACTACTT
	GFPdsR	taatacgactcactatagggGGAGTATTTTGTTGATAATGGTCTG
	*DcCHSA*iF	CGAGTGGTAGACCCGAACAT
	*DcCHSA*iR	ATGAAAAAGGCGAACACGAC
	*DcCHSA*dsF	taatacgactcactatagggCGAGTGGTAGACCCGAACAT
	*DcCHSA*dsR	taatacgactcactatagggATGAAAAAGGCGAACACGAC

**Table 2 insects-13-00728-t002:** Species and Gen Bank accession numbers of chitin synthase genes used in the phylogenetic tree construction.

Gene Name	GenBank Accession Number	Species
*DcCHSA*	this study	*Diaphorina citri*
*SeCHSA*	AAZ03545.1	*Spodoptera exigua*
*HaCHSA*	AKJ54482.1	*Helicoverpa armigera*
*EoCHS1*	ACA50098.1	*Ectropis obliqua*
*MsCHSA*	AAL38051.2	*Manduca sexta*
*PxCHS1*	API61827.1	*Plutella xylostella*
*DtCHS1*	XP_017009970.1	*Drosophila takahashii*
*CqCHSA*	XP_001866798.1	*Culex quinquefasciatus*
*AqCHS1*	ABD74441.1	*Anopheles quadrimaculatus*
*NvCHS1*	XP_001602290	*Nasonia vitripennis*
*BtCHSA*	XP_00339885	*Bombus terrestris*
*AmCHS1*	XP_395677	*Apis mellifera*
*AtCHS1*	AFM38193.1	*Anasa tristis*
*LhCHSA*	JAQ09912.1	*Lygus hesperus*
*AcCHSA*	KR611528.1	*Aphis citricidus*
*AgCHSA*	AFJ00066.1	*Aphis glycines*
*NlCHSA*	JQ040014	*Nilaparvata lugens*
*SfCHSA*	KY987034	*Sogatella furcifera*
*LsCHSA*	JQ040012	*Laodelphax striatellus*
*PcCHS1*	AJQ20794.1	*Panonychus citri*
*CmCHS2*	AJF93428.1	*Mamestra configurata*
*MysCHSB*	ASF79498.1	*Mythimna separate*
*SeCHSB*	ABI96087.1	*Spodoptera exigua*
*HaCHSB*	AKZ08595.1	*Helicoverpa armigera*
*DmCHSB*	CAC83725.1	*Drosophila melanogaster*

**Table 3 insects-13-00728-t003:** Toxicities of DFB to the fifth-instar nymph of *D. citri*.

Toxic Regression Equation	95% Confidence Interval	Correlation Coefficient	Chi-Square Value	Half Lethal Concentration (mg/L)
y = −4.012 + 1.867x	110.295–174.718	0.903	1.861	140.992

## Data Availability

The data generated during the study have already been reported in the manuscript.

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
