# Peer review of "Characterization of Chitin Synthase A cDNA from Diaphorina citri (Hemiptera: Liviidae) and Its Response to Diflubenzuron"

_insects, 2022, doi:10.3390/insects13080728_

Round 1
Reviewer 1 Report
This manuscript provides data on the chitin synthase A (DcCHSA) in Diaphorina citri, the effects of diflubenzuron (DFB) on D. citri nymph molting and DcCHSA transcription. The RNAi knockdown of CHS in D. citri was also investigated.
I feel this paper contains some useful data; however, other groups already performed the very similar work, and published a paper in 2019 (Lu ZJ, Huang YL, Yu HZ, Li NY, Xie YX, Zhang Q, et al. Silencing of the chitin synthase gene is lethal to the asian citrus psyllid, Diaphorina citri. Int J Mol Sci. 2019; https://www.mdpi.com/1422-0067/20/15/3734). They found that DcCHSA expressed throughout all developmental stages and tissues, had the highest expression level in the integument and fifth-instar nymph stage, they also investigated the effects of DFB on D. citri mortality and DcCHSA expression level. Also, at least two groups have shown that RNAi knockdown of CHS in D. citri causes increased lethality (Galdeano et al., 2017, Lu et al., 2019). It is very necessary to put these studies as references in this manuscript and discuss the results and provide readers what is the novelty here. By the form so far, it is hard to find out what is the new discoveries from this manuscript. The manuscript would be strengthened if the authors could point out some unique discoveries (compare with other similar studies). D. citri is a vector for the causative agents of Huanglongbing, which threatens citrus production worldwide. This article would benefit from a close editing, discussion, and clarification.
Specific recommendations for revision:
Summary
Page 1, line 10: what do you want to say? “Effects of diflubenzuron on D. citri nymph molting and CHS transcription”
Abstract
Page 1, line 18-23: nothing surprised, other group already performed the similar studies with the similar results (Lu et al., 2019), what is the novelty here?
Introduction
Page 2, line 53-54, based on the genome annotation from the research community, D. citri has one CHS gene (see the ref. Sherry Miller,Teresa D. Shippy,Blessy Tamayo,Prashant S. Hosmani,Mirella Flores-gonzalez,Lukas A. Mueller,Wayne B. Hunter,Susan J. Brown,Tom D’elia,Surya Saha, Annotation of chitin biosynthesis genes in Diaphorina citri, the Asian citrus psyllid,Gigabyte,2021 https://doi.org/10.46471/gigabyte.23)
Page 2, line 55-75, regarding to the effects of DFB and RNAi on CHS in D. citri, other groups already performed the studies (Lu et al., 2019, Galdeano et al., 2017, Tiwari et a., 2012). It is important to ref here and point out why you decide to perform the similar study.
Page 2, line 76-77, again other group reported the CHS with a complete ORF of 3180 bp from the genome database of D. citri (https://www.mdpi.com/1422-0067/20/15/3734. See the section of Identification of DcCHS and Bioinformatics Analysis). It needs to be referred, etc.
Materials and Methods
Page 2, line 97, for the five different concentrations of DFB, should not use ddH2O to make it, need to make sure the final concentration of the surfactant Triton X-100 should be 0.01% too
Page 5, line 176, could not access Table S1, not sure what are the CHS of the 20 insects
Results
Page 5, based on the previous reports (Lu et al., 2019), the ORF was 3180bp with 1059aa, here is 4437bp with 1478aa, big differences. Why? need to discuss it!
Page 5, Figure 1 needs to be modified and make it clear!
Page 7, line 207, what is 10?
Page 7, Figure 2 was a big mess: 1) for the complete amino acid sequence of D. citri CHSA, which one was used, from NCBI or the one listed in Figure 1? 2) 20 other insects, but based on the figure, it is 21 other insects! 3) Spodoptera exigua (Se) and Helicoverpa armigera (Ha), were repeated twice (see line 219, and 224-225), 4) where is Drosophila melanogaster (Dm) in the figure?
Page 9, line 258, the expression of the DcCHSA, not “the expression of the fifth instar nymph”
Discussion
The results were very similar as what previously reported by another group (Lu et al., 2019), from DcCHS expression profile, the effect of DFB on D. citri survival and DcCHS expression level, etc. Also, at least two groups have shown that RNAi knockdown of CHS in D. citri causes increased lethality (Galdeano et al., 2017, Lu et al., 2019). It is very necessary to discuss your results with the similar research and provide readers what is the novelty of your studies. Overall, the authors did not mention these papers, most of the references are based on the studies from aphid not D. citri.
Page 11, line 312-313, in terms of protein structure, Lu et al., (2019) studies showed the CHS protein in D. citri has 14 transmembrane helices, here is 15 … Again, it is important to discuss the differences here.
Page 11, line 321, “the presence of alternative splicing variants in DcCHSA”, put reference here
Reviewer 2 Report
The article published in the International Journal of Molecular Sciences (Lu ZJ, Huang YL, Yu HZ, Li NY, Xie YX, Zhang Q, Zeng XD, Hu H, Huang AJ, Yi L, Su HN. Silencing of the chitin synthase gene is lethal to the Asian citrus psyllid, Diaphorina citri. Int. J. Mol. Sci. 2019, 20, 3734; doi:10.3390/ijms20153734) has shown that DcCHSA is related to the epidermal formation of D. citri; knockdown of DcCHSA can disrupt the synthesis of the exo epidermal formation in D. citri, resulting in abnormal phenotype, reduced molting rate and increased mortality; therefore, DcCHSA can be used as a target for the prevention and control of D. citri.
The manuscript is highly similar to the published article (Lu et al., 2019). Therefore, the manuscript is not proper for publication.
Author Response
Our research is indeed somewhat similar to previous studies, but our results also have some highlights. First, we identified the complete sequence of CHS Gene of D. citri. The results of bioinformatics are more accurate than the previous results. Second, D. citri nymphs exposed to high DFB concentrations had a unique phenotype that was not seen in other research. After feeding dsRNA, we observed a stronger phenotype in nymphs and adults. In earlier research, the effect of RNAi were only seen in adults. As a result, our investigation produced some important findings. Thank you very much for your insightful advice.
Reviewer 3 Report
The manuscript by Zhang et al. (ID 1768851) reports the cloning of a cDNA for chitin synthase (DcCHSA) from Diaphorina citri, a pest of citrus crops and vector of citrus greening disease. Phylogenetic analysis revealed DcCHSA is most similar to the CHSA/CHS1 branch of insect chitin synthase proteins. The authors used RT-qPCR to examined the expression of DcCHSA during different developmental stages and tissue distribution and discovered it is most highly expressed during the 5th nymphal instar and in the head and integument. DcCHSA expression was increased in nymphs treated with diflubenzuron, an inhibitor of chitin synthesis. Knockdown of DcCHSA transcripts in 5th instar nymphs by RNAi resulted in increased mortality, lower rates of molting to adults, and failure to properly expand wings in adults that did molt. The authors propose that DcCHSA may be a suitable target for control strategies. The experiments are competently performed and the manuscript is well written. I have only minor suggestions for the authors.
1) Line 34: Please replace "inhaling" with "ingesting" or "feeding on" (inhaling involves respiration).
2) Line 132 (also line 255): For RT-qPCR, what is meant by "bacterial" tissue? Additionally, "intestine" (lines 132, 250 and 255) should be gut; the authors should also indicate if it is only midgut or if it also contains foregut and/or hindgut.
3) Line 185: Please replace "chitin synthase gene" with "chitin synthase transcript"
4) Line 193: "Figuer" should be "Figure".
5) Lines 190-197: Several references are made to supplemental figures (s1-s5) that were not provided; additionally, some of this information (transmembrane helices, putative N-glycosylation sites) is given in Fig. 1 and, therefore, is not necessary in the supplemental information.
6) Line 202: The legend to Fig 1 should read "The fifteen potential transmembrane helices were indicated in gray highlight ...". Additionally, the figure legend should also mention that the eleven potential N-glycosylation sites are boxed.
7) Line 211: "CHS genes" should be "CHS proteins"
8) Lines 219-225: In the figure legend it is not necessary to give the full species name for the insects used in the phylogenetic analysis as they are given in the figure.
9) Lines 230 - 232: "(number)-age nymph" should be replaced with"(number)-instar nymph"; also, "five" should be "fifth".
10) Line 237: Please describe how the current RT-qPCR results were different from previous studies and reference those studies here.
11) Section 3.4: Please describe how the phenotypes "mortality" and "abortive molting" differ. Do those that die after 48 h of DFB treatment include some individuals that molted to adult? Did the insects that failed to molt include both living and dead individuals or only living? For example, of the 31 % that did not molt after treatment with 500 mg/L DFB include all of the treated insects, or are they part of the 17% that did not die?
12) Section 3.5: The order of Fig. 5 and 6 should be switched since in the text the authors discuss the mortality and DcCHSA expression level first (Fig 6) and then the observed phenotypes of abortive molting or incomplete wing expansion (Fig 5).
13) Lines 292 and 297: "tail" should be replaced with "abdomen".
14) Line 308 and line 380: The description "CHS gene" should be "CHS cDNA".
15) Line 314: Please rephrase "had a higher genetic relationship with" to "was most similar to"; also, "CHS1 gene of brown citrus aphid" should be "CHS1 protein of brown citrus aphid".
16) Figure 1: Please insure that the print for the nucleotide and amino acid sequences are the same size for all of the figure. In the version reviewed the font size for the first two columns is much smaller than the last column; this gives the impression that they are two different figures (e.g. 1A and 1B) and that the authors are trying to emphasize different characteristics of the protein sequence.
Reviewer 4 Report
Please see the attached

Round 2
Author Response
请参阅附件。
